# Stiffness: A New Perspective on Generalization in Neural Networks

## Abstract

We investigate neural network training and generalization using the concept of stiffness. We measure how stiff a network is by looking at how a small gradient step on one example affects the loss on another example. In particular, we study how stiffness depends on 1) class membership, 2) distance between data points in the input space, 3) training iteration, and 4) learning rate. We experiment on MNIST, FASHION MNIST, and CIFAR-10 using fully-connected and convolutional neural networks. Our results demonstrate that stiffness is a useful concept for diagnosing and characterizing generalization. We observe that small learning rates reliably lead to higher stiffness at a given epoch as well as at a given training loss. In addition, we measure how stiffness between two data points depends on their mutual input-space distance, and establish the concept of a dynamical critical length that characterizes the distance over which datapoints react similarly to gradient updates. The dynamical critical length decreases with training and the higher the learning rate, the smaller the critical length.

## 1 Introduction

Neural networks are a class of highly expressive function approximators that proved to be successful in approximating solutions to complex tasks across many domains such as vision, natural language understanding, and game-play. They have long been recognized as universal function approximators (Hornik et al., 1989; Cybenko, 1989; Leshno et al., 1993). The specific details that lead to their expressive power have recently been studied in Montúfar et al. (2014); Raghu et al. (2017); Poole et al. (2016). Empirically, neural networks have been extremely successful at generalizing to new data despite their over-parametrization for the task at hand, as well as their proven ability to fit arbitrary random data perfectly Zhang et al. (2016); Arpit et al. (2017).

The fact that gradient descent is able to find good solutions given the highly over-parametrized family of functions has been studied theoretically in Arora et al. (2018) and explored empirically in Li et al. (2018), where the effective low-dimensional nature of many common learning problems is shown. Fort & Scherlis (2019) extends the analysis in Li et al. (2018) to demonstrate the role of initialization on the effective dimensionality, and Fort & Jastrzebski (2019) use the result to build a phenomenological model of the loss landscape.

Du et al. (2018a) and Du et al. (2018b) use a Gram matrix to study convergence in neural network empirical loss. Pennington & Worah (2017) study the concentration properties of a similar covariance matrix formed from the output of the network. Ghorbani et al. (2019) investigate the Hessian eigenspectrum and Papyan (2019) show how it is related to the gradient covariance. Both concepts are closely related to our definition of stiffness.

To explain the remarkable generalization properties of neural networks, it has been proposed (Rahaman et al., 2018) that the function family is biased towards low-frequency functions. The role of similarity between the neural network outputs to similar inputs has been studied in Schoenholz et al. (2016) for random initializations and explored empirically in Novak et al. (2018).

### 1.1 Our contribution

In this paper, we study generalization through the lens of *stiffness*. We measure how *stiff* a neural network is by analyzing how a small gradient step based on one input example affects the loss

on another input example. Mathematically, if the gradient of the loss at point $X_1$ with respect to the network weights is $\nabla_W \mathcal{L}(X_1) = \vec{g}_1$, and the gradient at point $X_2$ is $\vec{g}_2$, we define stiffness $\propto \vec{g}_1 \cdot \vec{g}_2$. We specifically focus on the sign of $\vec{g}_1 \cdot \vec{g}_2$ as well as the cosine between the two vectors $\cos(\vec{g}_1, \vec{g}_2) = \hat{g}_1 \cdot \hat{g}_2$, where $\hat{g} = \vec{g}/|\vec{g}|$, which both capture the resistance of the functional approximation learned to deformation by gradient steps. We find the concept of stiffness useful in diagnosing and characterizing generalization. As a corollary, we use stiffness to characterize the regularization power of learning rate, and show that higher learning rates bias the functions learned towards lower stiffness.

We show that stiffness is directly related to generalization, and in particular that it starts dropping sharply at the moment training and validation set losses stop evolving together. We explore the concept of stiffness for fully-connected (FC) and convolutional neural networks (CNN) on 3 classification datasets (MNIST, FASHION MNIST, CIFAR-10). We focus on how stiffness between data points depends on their 1) class membership, 2) distance between each other in the space of inputs, 3) training epoch, and 4) the choice of learning rate. We study stiffness between two images in the training set, one in the training and one in the validation set, and both images in the validation set.

We observed the stiffness based on class membership and noticed a clear evolution towards higher stiffness for images between different classes and towards lower stiffness within the same class. We diagnose and characterize the class-dependent stiffness matrix for fully-connected and convolutional neural networks on the datasets mentioned above in different stages of training. We observe the stiffness between inputs to regress to zero with the onset of overfitting, demonstrating the clear connection to generalization.

The choice of learning rate effects the stiffness properties of the learned function significantly. High learning rates induce functional approximations that are less stiff over larger distances (i.e. data points further apart stop responding similarly to gradient updates). We define the concept of *dynamical critical length* to capture the phenomenon.

This paper is structured as follows: we introduce the concept of stiffness and the relevant theory in Section 2. We describe our experimental setup in Section 3, and discuss their results in Section 4. We conclude with Section 5.

## 2 THEORETICAL BACKGROUND

### 2.1 STIFFNESS – DEFINITIONS

Let a functional approximation (e.g. a neural network) $f$ be parametrized by tunable parameters $W$. Let us assume a classification task and let a data point $X$ have the ground truth label $y$. A loss $\mathcal{L}(f_W(X), y)$ gives us the amount of mismatch between the function's output at input $X$ and the ground truth $y$. The gradient of the loss with respect to the parameters

$$\vec{g} = \nabla_W \mathcal{L}(f_W(X), y) \tag{1}$$

is the direction in which, if we were to change the parameters $W$, the loss would change the most rapidly (for infinitesimal step sizes). Gradient descent uses this step (the negative of it) to update the weights and gradually tune the functional approximation to better correspond to the desired outputs on the training dataset inputs. Let us consider two data points with their ground truth labels $(X_1, y_1)$

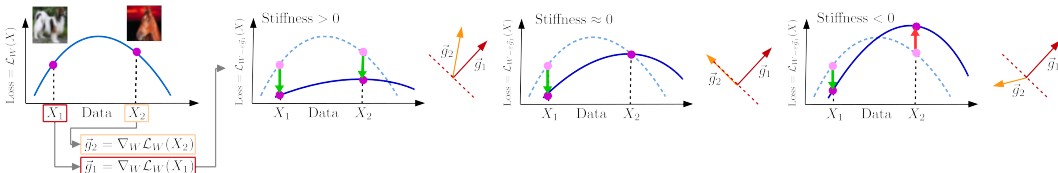

Figure 1: A diagram illustrating the concept of stiffness. It can be viewed in two equivalent ways: a) as the change in loss at a datapoint induced by the application of a gradient update based on another datapoint, and b) the alignment of loss gradients computed at the two datapoints.

and $(X_2, y_2)$. We construct a gradient with respect to example 1 as $\vec{g}_1 = \nabla_W \mathcal{L}(f_W(X_1), y_1)$ and

ask how do the losses on data points 1 and 2 change as the result of a small change of $W$ in the direction $-\vec{g}_1$, i.e. what is

$$\Delta\mathcal{L}_1 = \mathcal{L}(f_{W-\varepsilon\vec{g}_1}(X_1), y_1) - \mathcal{L}(f_W(X_1), y_1), \tag{2}$$

which is equivalent to

$$\Delta\mathcal{L}_1 = -\varepsilon\nabla_\varepsilon\mathcal{L}(f_{W-\varepsilon\vec{g}_1}(X_1), y_1) = -\varepsilon\vec{g}_1 \cdot \vec{g}_1. \tag{3}$$

The change in loss on input 2 due to the same gradient step from input 1 becomes equivalently $\Delta\mathcal{L}_2 = -\varepsilon\nabla_\varepsilon\mathcal{L}(f_{W-\varepsilon\vec{g}_1}(X_2), y_2) = -\varepsilon\vec{g}_1 \cdot \vec{g}_2$ We are interested in the correlation in loss changes $\Delta\mathcal{L}_1$ and $\Delta\mathcal{L}_2$. We know that $\Delta\mathcal{L}_1 < 0$ since we constructed the gradient update accordingly. We define positive stiffness to mean $\Delta\mathcal{L}_2 < 0$ as well, i.e. that losses at both inputs went down. We assign the stiffness of 0 for $\Delta\mathcal{L}_2 = 0$. If $\Delta\mathcal{L}_2 > 0$, the two inputs would be anti-stiff (negative stiffness). The equations above show that this can equivalently be thought of as the overlap between the two gradients $\vec{g}_1 \cdot \vec{g}_2$ being positive for positive stiffness, and negative for negative stiffness. We illustrate this in Figure 1.

The above indicate that what we initially conceived of as a change in loss due to the application of a small gradient update from one input to another is in fact equivalent to analyzing *gradient alignment* between different datapoints.

We will be using 2 different definitions of stiffness: the sign stiffness and the cosine stiffness. We define the *sign* stiffness to be the expected sign of $\vec{g}_1 \cdot \vec{g}_2$ (or equivalently the expected sign of $\Delta\mathcal{L}_1\Delta\mathcal{L}_2$) as

$$S_{\text{sign}}((X_1, y_1), (X_2, y_2); f) = \mathbb{E}\left[\text{sign}\left(\vec{g}_1 \cdot \vec{g}_2\right)\right], \tag{4}$$

where stiffness depends on the dataset from which $X_1$ and $X_2$ are drawn. The cosine stiffness is

$$S_{\cos}((X_1, y_1), (X_2, y_2); f) = \mathbb{E}\left[\cos\left(\vec{g}_1 \cdot \vec{g}_2\right)\right], \tag{5}$$

where $\cos\left(\vec{g}_1 \cdot \vec{g}_2\right) = (\vec{g}_1/|\vec{g}_1|) \cdot (\vec{g}_2/|\vec{g}_2|)$. We use both versions of stiffness as they are suitable to highlight different phenomena – the *sign* stiffness shows the stiffness between classes clearer, while the *cosines* stiffness is more useful for within-class stiffness.

## 2.2 TRAIN-TRAIN, TRAIN-VAL, AND VAL-VAL

When measuring stiffness between two datapoints, we have 3 options: 1) choosing both datapoints from the training set (we call this *train-train*), 2) choosing one from the training set and the other from the validation set (*train-val*), and 3) choosing both from the validation set (*val-val*). The train-val stiffness is directly related to generalization, as it corresponds to the amount of improvement on the training set transferring to the improvement of the validation set. We empirical observe that all 3 options are empirically remarkably similar, which gives us confidence that they all track generalization. This is further supported by observing their behavior is a function of epoch in Figure 2.

## 2.3 STIFFNESS BASED ON CLASS MEMBERSHIP

A natural question to ask is whether a gradient taken with respect to an input $X_1$ in class $c_1$ will also decrease the loss for example $X_2$ with true class $c_2$. In particular, we define the *class stiffness matrix*

$$C(c_a, c_b) = \mathop{\mathbb{E}}_{X_1 \in c_a, X_2 \in c_b}\left[S((X_1, y_1), (X_2, y_2))\right]. \tag{6}$$

The on-diagonal elements of this matrix correspond to the suitability of the current gradient update to the members of a class itself. In particular, they correspond to *within class* generalizability. The off-diagonal elements, on the other hand, express the amount of improvement transferred from one class to another. They therefore directly diagnose the amount of generality the currently improved features have.

A consistent summary of generalization between classes is the off-diagonal sum of the class stiffness matrix

$$S_{\text{between classes}} = \frac{1}{N_c(N_c - 1)} \sum_{c_1} \sum_{c_2 \neq c_1} C(c_1, c_2). \tag{7}$$

In our experiments, we track this value as a function of learning rate once we reached a fixed loss. The quantity is related to how generally applicable the learned features are, i.e. how well they transfer from one class to another. For example, for CNNs learning good edge detectors in initial layers typically benefits all downstream tasks, regardless of the particular class in question. We do the equivalent for the within-class stiffness (= on diagonal elements). When the within-class stiffness starts going $< 1$, the generality of the features improved does not extend to even the class itself.

### 2.4 STIFFNESS AS A FUNCTION OF DISTANCE

We investigate how stiff two inputs are based on how far away from each other they are. We can think of neural networks as a form of kernel learning and here we are investigating the particular form of the learned kernel. This links our results to the work on spectral bias (towards slowly-varying, low frequency functions) in Rahaman et al. (2018). We are able to directly measure the characteristic size of the stiff regions in neural networks trained on real tasks which we call *dynamical critical length* $\xi$. We work with data normalized to the unit sphere $|\vec{X}| = 1$ and use their mutual cosine to define their distance as

$$\text{distance}(\vec{X_1}, \vec{X_2}) = 1 - \frac{\vec{X_1} \cdot \vec{X_2}}{|\vec{X_1}||\vec{X_2}|} . \tag{8}$$

which has the advantage of being bounded between 0 and 2. We track this threshold distance $\xi$ as a function of training and learning rate to estimate the characteristic size of the stiff regions of a neural net.

## 3 METHODS

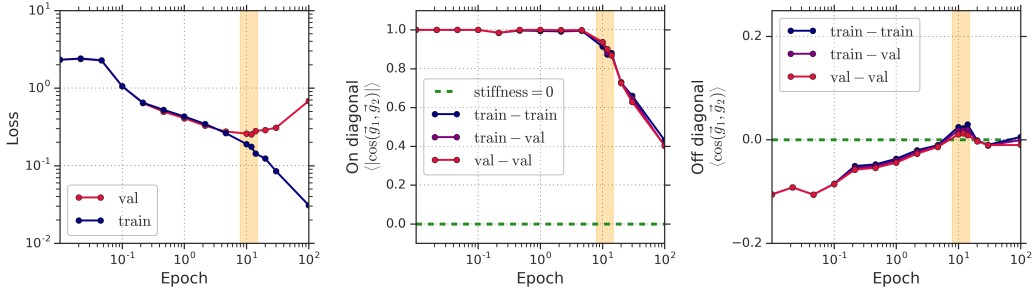

Figure 2: The evolution of training and validation loss (*left panel*), within-class stiffness (*central panel*) and between-classes stiffness (*right panel*) during training. The onset of over-fitting is marked in orange. After that, both within-class and between-classes stiffness regress to 0. The same effect is visible in stiffness measured between two training set datapoints, one training and one validation datapoint, and two validation set datapoints.

### 3.1 EXPERIMENTAL SETUP

We ran a large number of experiments with fully-connected (FC) and convolutional neural networks (CNN) on 3 classification datasets: MNIST (LeCun & Cortes, 2010), FASHION MNIST Xiao et al. (2017), and CIFAR-10 Krizhevsky (2009). Using those experiments, we investigated the behavior of stiffness as a function of 1) training epoch, 2) the choice of learning rate, 3) class membership, and 4) the input space distance between images.

For experiments with fully-connected neural networks, we used a 3 layer ReLU network of the form $X \rightarrow 500 \rightarrow 300 \rightarrow 100 \rightarrow y$. For experiments with convolutional neural networks, we used a 3 layer network with filter size 3 and the numbers of channels being 16, 32, and 32 after the respective convolutional layer, each followed by $2 \times 2$ max pooling. The final layer was fully-connected. No batch normalization was used.

We pre-processed the network inputs to have zero mean and unit variance, and normalized all data to the unit sphere as $|\vec{X}| = 1$. We used $\mathrm{Adam}$ with different (constant) learning rates as our optimizer and the default batch size of 32.

### 3.2 TRAINING AND STIFFNESS EVALUATION

We evaluated stiffness properties between data points from the training set, one from the training and one from the validation set, and both from the validation set. We used the training set to train our model. The procedure was as follows: 1) Train for a number of steps on the *training* set and update the network weights accordingly. 2) For each of the modes { train-train, train-val, and val-val}, go through tuples of images coming from the respective datasets. 3) For each tuple calculate the loss gradients $g_1$ and $g_2$, and compute check $\mathrm{sign}(\vec{g}_1 \cdot \vec{g}_2)$ and $\cos(\vec{g}_1, \vec{g}_2)$. 4) Log the input space distance between the images as well as other relevant features. In our experiments, we used a fixed subset (typically of $\approx 500$ images for experiments with 10 classes) of the training and validation sets to evaluate the stiffness properties on. We convinced ourselves that such a subset is sufficiently large to provide measurements with small enough statistical uncertainties, which we overlay in our figures.

### 3.3 LEARNING RATE DEPENDENCE

We investigated how stiffness properties depend on the learning rate used in training. To be able to do that, we first looked at the dynamical critical scale $\xi$ for each training step of our network, and then compared those based on the epoch and training loss at the time in order to be able to compare training runs with different learning rates fairly. The results are shown in Figures 4 and 6.

## 4 RESULTS

### 4.1 STIFFNESS PROPERTIES BASED ON CLASS MEMBERSHIP

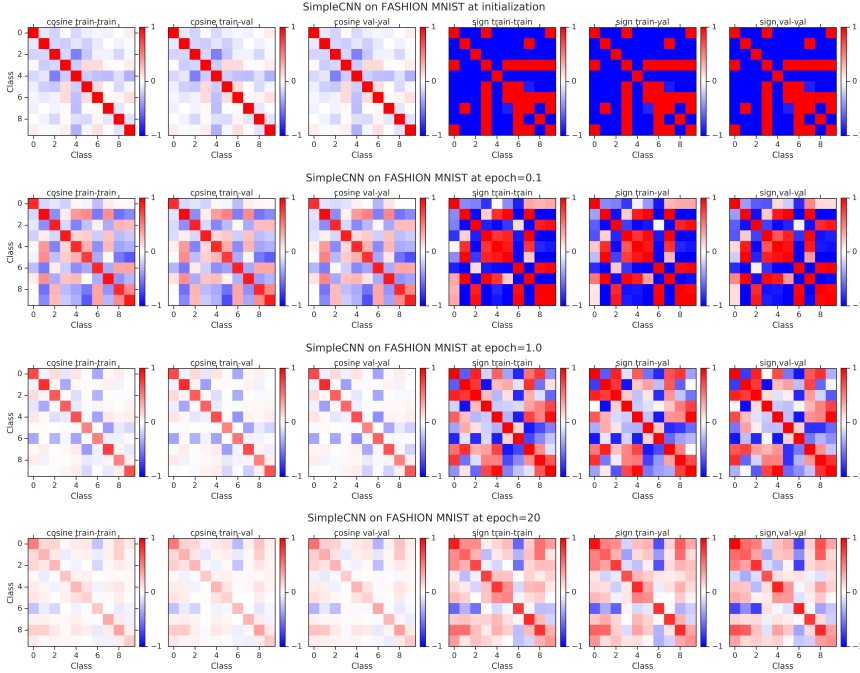

Figure 3: Class membership dependence of stiffness for a CNN on FASHION MNIST at 4 different stages of training. The figure shows stiffness between train-train, train-val and val-val pairs of images, as well as the sign and cosine metrics.

We explored the stiffness properties based on class membership as a function of training epoch at 4 different stages of training – at initialization, very early, around epoch 1, and at the late stage. Our results are summarized in Figures 3 and Figures 7 and 8. The within-class (on diagonal) and between-classes results are summarized in Figure 4, which as an example of equivalent plots we generated for all our experiments. Initially, an improvement based on an input from a particular

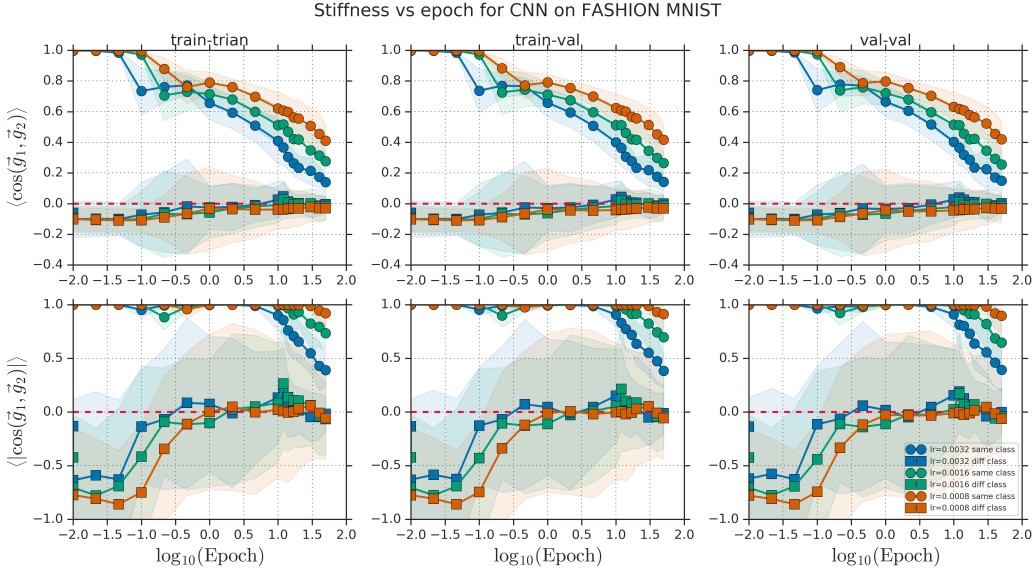

Figure 4: Stiffness as a function of epoch. The plots summarize the evolution of within-class and between-classes stiffness measures as a function of epoch of training for a CNN on FASHION MNIST.

class benefits only members of the same class. Intuitively, this could be due to some crude features shared within a class (such as the typical overall intensity, or the average color) being learned. There is no consistent stiffness between different classes at initialization. As training progresses, within-class stiffness stays high. In addition, stiffness between classes increases as well, given the model were is powerful enough for the dataset. With the onset of overfitting, as shown in Figure 2, the model becomes increasingly less stiff until even stiffness for inputs within the same class is lost.

## 4.2 STIFFNESS AS A FUNCTION OF DISTANCE BETWEEN DATAPOINTS

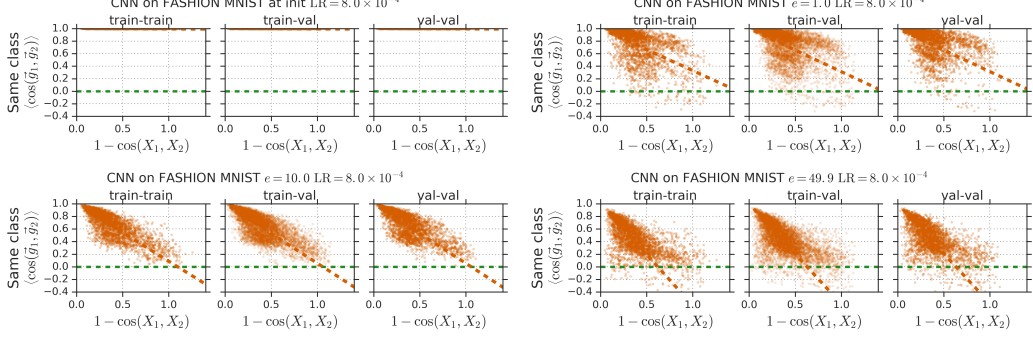

Figure 5: Stiffness between images of the same class as a function of their input space distance for 4 different stages of training of a CNN on FASHION MNIST.

We investigated stiffness between two inputs as a function of their distance in the input space in order to measure how large the patches of the learned function that move together under gradient

updates are. We focused on examples from the same class. Examples of our results are shown in Figure 5 and Figures 9 and 10. Fitting a linear function to the data, we estimate the distance at which stiffness goes to 0, and called it the *dynamical critical length $\xi$*. Equivalent plots were generated for each epoch of training in each of our experiments in order to measure $\xi$ used in Figure 6.

## 4.3 THE DYNAMICAL CRITICAL LENGTH $\xi$, AND THE ROLE OF LEARNING RATE

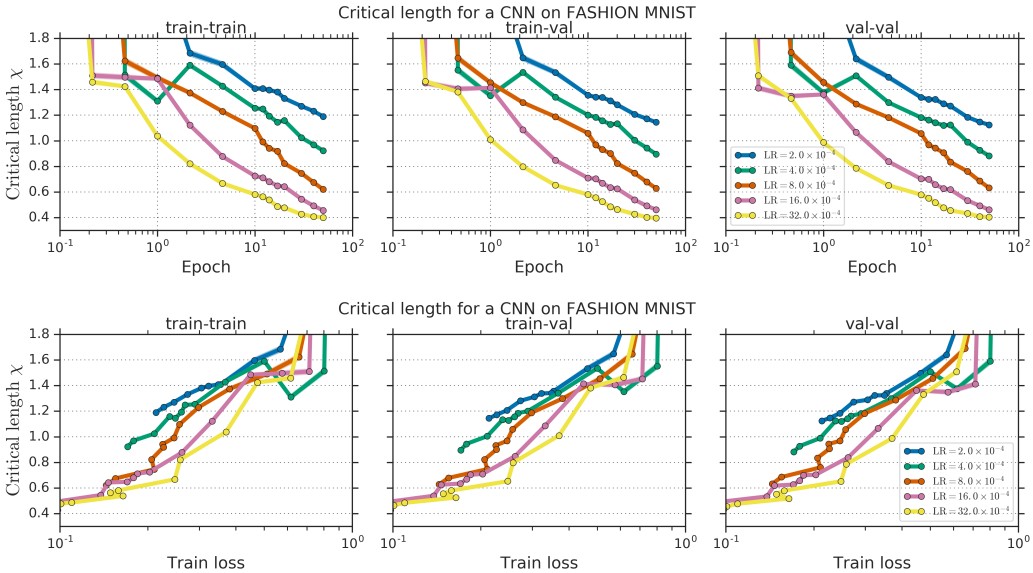

Figure 6: The effect of learning rate on the stiffness dynamical critical scale $\xi$ – the input space distance between images over which stiffness disappears. The upper part shows $\xi$ for 5 different learning rates as a function of epoch, while the lower part shows it as a function of training loss in order to be able to compare different learning rates fairly. The larger the learning rate, the smaller the stiff domains.

At each epoch of training for each of our experiments, we analyzed the distribution of within-class stiffness between images based on their distance, and extracted the zero crossing which we call the critical dynamical scale $\xi$. In Figures 6 and 11 we summarize the dependence of $\xi$ on the epoch of training as well as the training loss for 5 different learning rates. We use the training loss to make sure we are comparing runs with different learning rates at the equivalent stages of training. We see that the bigger the learning rate, the smaller the domain size $\xi$.

## 4.4 STIFF DOMAIN SIZE AS THE CHARACTERISTIC LENGTH SCALE?

A natural question arises as to whether the characteristic distance between two input points at which stiffness reaches zero defines the typical scale of spatial variation of the learned function. Unfortunately, that is *not* necessarily the case, though it can be for some families of functions. The stiff domain sizes visible in e.g. Figure 5 represent the typical length scale over which neural networks *react* similarly to gradient inputs, rather than the typical length scale of variation of the function value itself.

To illustrate the difference, imagine a function that varies rapidly over input data, however, whose losses over the same data move in the same direction on application of a gradient step based on any of the data points. This function would have a small characteristic length scale of value variation, yet large stiff domain size.

## 5  DISCUSSION AND CONCLUSION

We explored the concept of neural network stiffness and used it to diagnose and characterize generalization. We studied stiffness for models trained on real datasets, and measured its variation with training iteration, class membership, distance between data points, and the choice of learning rate. We explored stiffness between pairs of data points coming both from the training set, one from the training and one from the validation set, and both from the validation set. Training-validation stiffness is directly related to the transfer of improvements on the training set to the validation set. We used two different stiffness metrics – the sign stiffness and the cosine stiffness – to highlight different phenomena.

On real data, we explored models trained on MNIST, FASHION MNIST, and CIFAR-10 through the lens of stiffness. In essence, stiffness measures the alignment of gradients taken at different input data points, which we show is equivalent to asking whether a weight update based on one input will benefit the loss on another. We demonstrate the connection between stiffness and generalization and show that with the onset of overfitting to the training data, stiffness decreases and eventually reaches 0, where even gradient updates taken with respect images of a particular class stop benefiting other members of the same class. This happens even within the training set itself, and therefore could potentially be used as a diagnostic for early stopping to prevent overfitting.

Having established the usefulness of stiffness as a diagnostic tool for generalization, we explored its dependence on class membership. We find that in general gradient updates with respect to a member of a class help to improve loss on data points in the same class, i.e. that members of the same class have high stiffness with respect to each other. This holds at initialization as well as throughout most of the training. The pattern breaks when the model starts overfitting to the training set, after which within-class stiffness eventually reaches 0. We observe this behavior with fully-connected and convolutional neural networks on MNIST, FASHION MNIST, and CIFAR-10.

Stiffness between inputs from different classes relates to the generality of the features being learned and within-task transfer of improvement from class to class. With the onset of overfitting, the stiffness between different classes regresses to 0, as does within-class stiffness.

We also investigated the characteristic size of stiff regions in our trained networks at different stages of training. By studying stiffness between two inputs and measuring their distance in the input space, we observed that the farther the datapoints and the higher the epoch of training, the less stiffness exists between them on average. This allowed us to define the *dynamical critical scale $\xi$* – an input space distance over which stiffness between input points decays to 0. $\xi$ corresponds to the size of stiff regions – patches of the data space that can move together when a gradient update is applied, provided it were an infinitesimal gradient step. For finite step sizes, the matter becomes more complicated, as the linear regime in which we operate ceases to apply.

We investigated the effect of learning rate on stiffness by observing how $\xi$ changes as a function of epoch and the training loss for different learning rates. We show that the higher the learning rate, the smaller the $\xi$, i.e. for high learning rates the patches of input space that are improved together are smaller. This holds both as a function of epoch and training loss, which we used in order to compare runs with different learning rates fairly. This points towards the regularization role of learning rate on the kind of function we learn. We observe significant differences in the characteristic size of regions of input space that react jointly to gradient updates based on the learning rate used to train them.

In this paper, all the experiments were conducted with two fixed architectures. One obvious extension to the concept of stiffness would be to ascertain the role stiffness might play in architecture search. For instance, we expect locality (as in CNN) to reflect in higher stiffness properties. It is quite possible that stiffness could be a guiding parameter for meta-learning and explorations in the space of architectures, however, this is beyond the scope of this paper and a potential avenue for future work.

In summary, we defined the concept of stiffness, showed its utility in providing a perspective to better understand generalization characteristics in a neural network and observed its variation with learning rate.

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

# A APPENDIX

## A.1 ADDITIONAL CLASS STIFFNESS MATRIX RESULTS

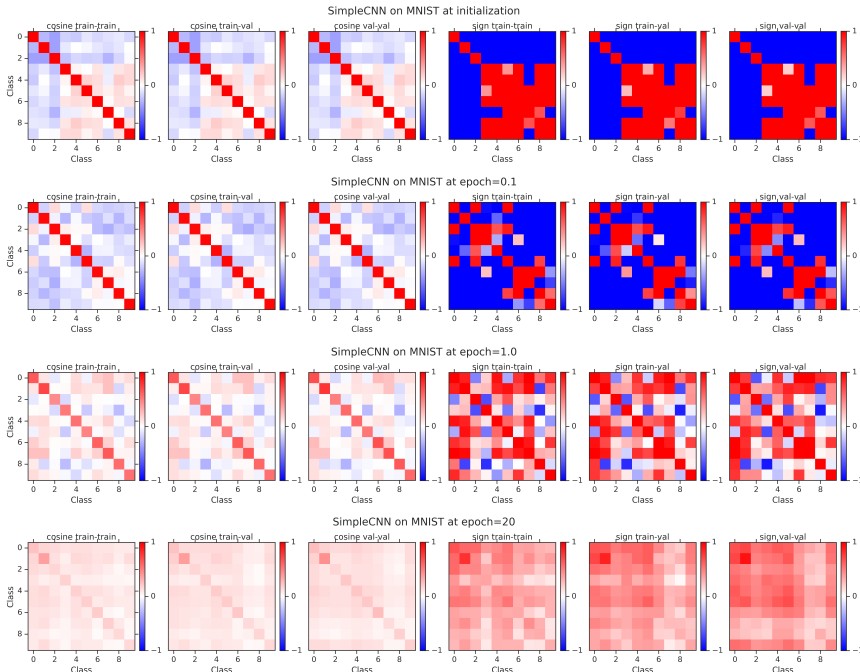

Figure 7: Class-membership dependence of stiffness for a CNN on MNIST at 4 different stages of training.

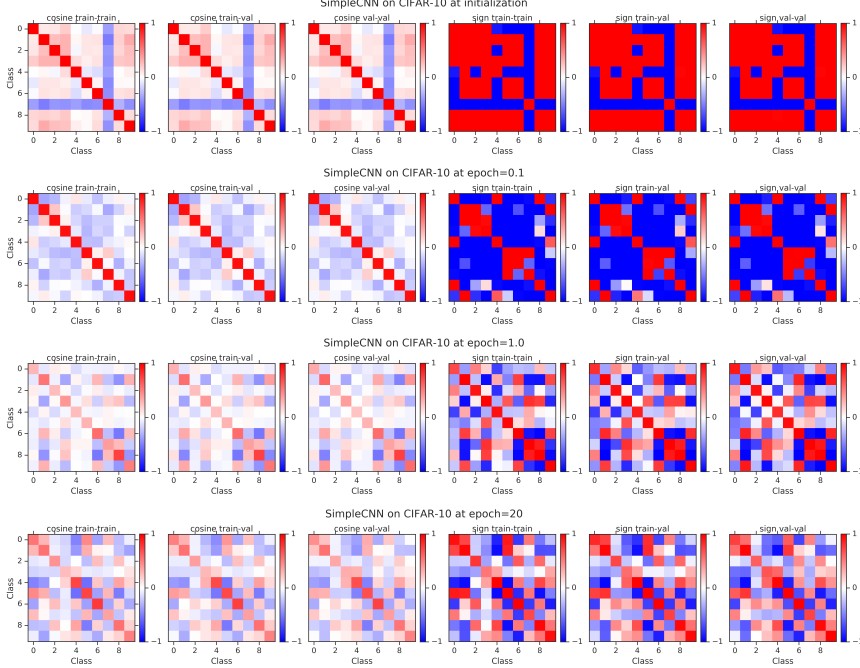

Figure 8: Class-membership dependence of stiffness for a CNN on CIFAR-10 at 4 different stages of training.

## A.2 ADDITIONAL STIFFNESS AS A FUNCTION OF DATAPOINT SEPARATION RESULTS

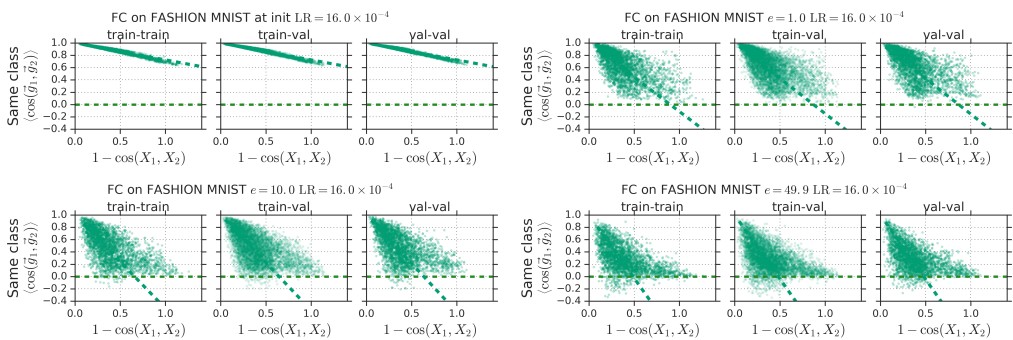

Figure 9: Stiffness between images of the same class as a function of their input space distance for 4 different stages of training of a fully-connected network (FC) on FASHION MNIST.

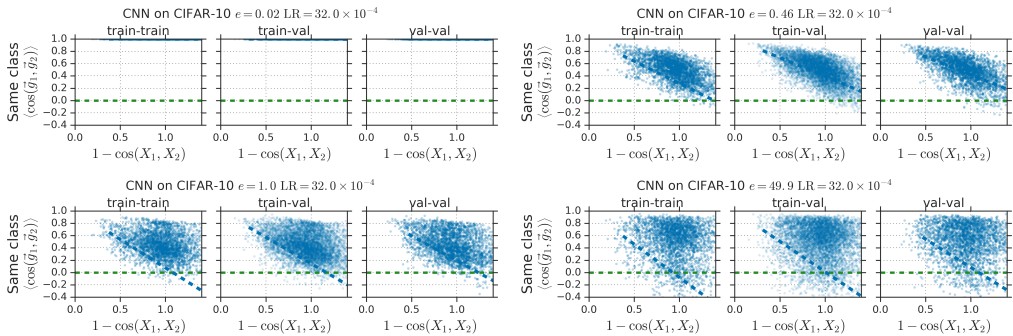

Figure 10: Stiffness between images of the same class as a function of their input space distance for 4 different stages of training of a CNN on CIFAR-10.

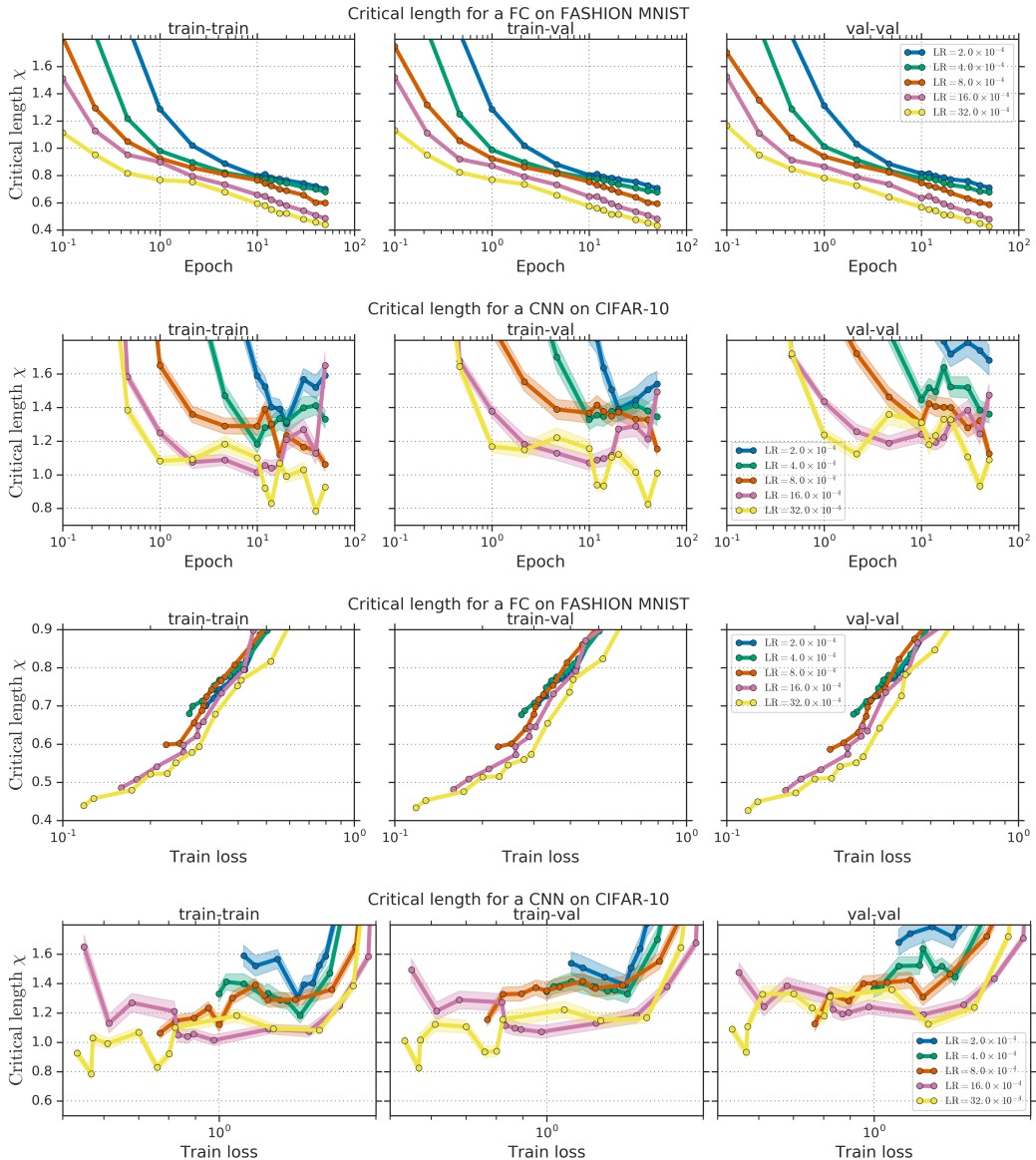

Figure 11: The effect of learning rate on the stiffness dynamical critical scale $\xi$ – the input space distance between images over which stiffness disappears. The upper part shows $\xi$ for 5 different learning rates as a function of epoch, while the lower part shows it as a function of training loss in order to be able to compare different learning rates fairly. The larger the learning rate, the smaller the stiff domains.

