# OpenReview forum: "Stiffness: A New Perspective on Generalization in Neural Networks"
_ICLR.cc/2020/Conference — Reject_

### Official Review · AnonReviewer1 · 2019-10-21
**Official Blind Review #1**

**Rating:** 6

**Review:**

This submission introduces a metric, termed stiffness, to evaluate the generalization capability of neural networks. The metric is novel and straightforward, it measures how stiff a network is by looking at how a small gradient step on one example affects the loss on another example. The authors study several configurations  on three small datasets. They demonstrate that stiffness is a useful concept for diagnosing and characterizing generalization.

I give an initial rating of weak accept because (1) The paper is well motivated and well written. Studying generalization is important for neural networks. (2) It seems from experiments that stiffness is a useful metric to indicate models' generalization capability. However, I have a few concerns.

First, the authors study several configurations like train-train, train-val and val-val. However, these configurations are still in-domain analysis, the data distribution is quite similar. It can not support author's claims well. Adding an experiment where domain gap is large will make the submission stronger, such as train-test, cross-dataset or challenging tasks like semantic segmentation.

Second, the datasets being used are very small. I understand that for theoretically analysis, small datasets are quick to converge and easy to demonstrate. However, this submission focuses on generalization problem during transfer learning. Hence, it needs at least a bigger dataset, like ImageNet, to show it really works.

**Experience Assessment:**

I have read many papers in this area.

**Review Assessment: Checking Correctness Of Derivations And Theory:**

I assessed the sensibility of the derivations and theory.

**Review Assessment: Checking Correctness Of Experiments:**

I assessed the sensibility of the experiments.

**Review Assessment: Thoroughness In Paper Reading:**

I read the paper at least twice and used my best judgement in assessing the paper.

---

> ### Author Response · Authors · 2019-11-15
> **Response to Review #1**
>
> Thank you for your review and comments. We hope that you will champion our paper.
>
> We provide a detailed response to the points you brought up below.
>
> ------------
> “First, the authors study several configurations like train-train, train-val and val-val. However, these configurations are still in-domain analysis, the data distribution is quite similar.”
>
> We studied the stiffness between input images from the training set and the validation set precisely in order to directly quantify the transfer of performance improvement gained on the training set to the unseen validation set. We agree that those distributions are hopefully very similar, however, they are not identical. In that sense, this constitutes a weak version of the out-of-distribution performance experiment you were suggesting.
>
> We agree that adding an experiment where the domain gap is large might be interesting, however, the focus of our paper was not on transfer learning (where this is a common regime), but rather on learning on a specific dataset itself. The transfer we were concerned with was from one example to another, i.e. within dataset generalization. If we, for example, looked at the stiffness between train images and random noise images, the interpretation of that metric would be very difficult, as it is not a priori known what kind of behavior would even be desirable there. It could even be the case that you do not want to transfer any performance to random out-of-distribution images, as this could limit your performance on the actual distribution.
>
> ---------------------
> “Second, the datasets. I understand that for theoretically analysis, small datasets are quick to converge and easy to demonstrate. However, this submission focuses on generalization problem during transfer learning. Hence, it needs at least a bigger dataset, like ImageNet, to show it really works.”
>
> We understand that looking at ImageNet would strengthen our case, however, the stiffness calculations are very computationally demanding and the consistent appearance of the effects on MNIST, Fashion MNIST and CIFAR-10 is, according to us, a good indication of their generality. In addition, the goal of our paper is to introduce a metric and show its usefulness, for which we believe CIFAR-10 can be sufficient. Nonetheless, we will try to show our results on larger datasets.

---

### Official Review · AnonReviewer4 · 2019-11-05
**Official Blind Review #4**

**Rating:** 3

**Review:**

This paper introduces the concept of stiffness: a measure of the change in the loss of sample A due to a gradient step based on sample B. It analyses the expected dynamic for A, B samples from the same and different classes, as well as, samples from the train and test sets.

To better understand the dynamics of optimization in neural networks is an open and important problem and the paper is clearly motivated in this regard. The proposed method is straight forward and I am not aware of a similar method.

In addition to that, the paper also introduces "dynamical critical length ξ" which is the stiffness of A, B samples based on the cosine similarity of the respective inputs (section 2.4). A linear estimator of when this length becomes 0 is also introduced. Confusingly this is also called the "dynamical critical length ξ" in section 4.2. Later on the term "dynamical scale ξ" and "dynamical critical scale ξ" seem to be used interchangeably. Figure 6 mentions the "critical length χ" on the y-axis which seems to be a typo as no such measure was introduced.

The equivalence between eq. 2 and the two parts of eq. 3 is not obvious. We'd appreciate if the authors would provide a proof of such.

Overall, the paper is written in a simple language but paragraphs remain surprisingly hard to understand. An example of such is e.g. section 4.4: What do the authors mean by "characteristic distance" between two input points? What is "the typical scale of spatial variation" of a function? etc.

The paper concludes that:

1.) there is a link between generalization and stiffness
2.) stiffness decreases with the onset of overfitting
3.) "general gradient updates with respect to a member of a class help to improve loss on data points in the same class"
4.) "The pattern breaks when the model starts overfitting to the training set, after
which within-class stiffness eventually reaches 0"
5.) This is observed for different models on different datasets
6.) "we observed that the farther the datapoints and the higher the epoch of training, the less
stiffness exists between them on average"
7.) "the higher the learning rate, the smaller the ξ"

Verdict: Reject

The conclusions are self-evident. The paper fails to demonstrate the usefulness of stiffness and most results are expected and provide little to no insights into the optimization dynamics of deep neural networks. In fact, the reasoning in this paper is almost tautological (conclusions 1-6).

E.g. if the A, B samples used to compute stiffness are separately drawn from the train and test set then stiffness is a proxy for the difference between the train error and the test error after another gradient step. The authors then compute stiffness at different points of the optimization procedure and conclude that stiffness decreases when the network starts to overfit. Since overfitting is the point in training where train error and test error diverge it is obvious that this can also be observed with regards to "stiffness". Hence, the reasoning is circular.

Conclusion 7 is slightly different in that it observes that larger learning rates result in smaller ξ which, given the previous paragraph, we can rewrite into the statement "larger learning rates generalise better". This is a well known empirical observation and has been discussed thoroughly (e.g. on connection with flat and sharp minima or learning rate decay schedules).

Disclaimer: This review was done on short notice.

**Experience Assessment:**

I have read many papers in this area.

**Review Assessment: Checking Correctness Of Derivations And Theory:**

N/A

**Review Assessment: Checking Correctness Of Experiments:**

I assessed the sensibility of the experiments.

**Review Assessment: Thoroughness In Paper Reading:**

I read the paper thoroughly.

---

> ### Author Response · Authors · 2019-11-15
> **Response to Review #4  1/2**
>
> Thank you for your review and comments. We appreciate that your review was done on a very short notice and thank you for it. We would like to address and dispute several of the points your brought up.
>
> ---------------------------
> “Overall, the paper is written in a simple language but paragraphs remain surprisingly hard to understand. An example of such is e.g. section 4.4: What do the authors mean by "characteristic distance" between two input points?” …. “ What is "the typical scale of spatial variation" of a function?”
>
> We would like to clarify the points of confusion in Subsection 4.4. The “characteristic distance” point you bring up is a part of a longer statement:
> “A natural question arises as to whether the characteristic distance between two input points at which stiffness reaches zero defines the typical scale of spatial variation of the learned function.”
> So the first “characteristic distance” refers to the distance between two points in the input space where the gradient step with respect to one of them will not influence the other one. This is the quantity we call "dynamical critical length/scale ξ" and which we empirically measure in real networks trained on real data in Figures 5, 6, 9, 10 and 11.
>
> The "the typical scale of spatial variation" of a function is its dominant Fourier mode in the input space -- i.e. the length scale in the input space over which the predictions change significantly. This scale is related to the concept often called the spectral bias of neural networks and there is a large literature on the topic.
>
> Our point in Subsection 4.4 was to make clear that the scale on which input points do not influence each other under gradient updates = "dynamical critical length/scale ξ", and the scale over which the predictions change significantly in the input space = "the typical scale of spatial variation", are not necessarily the same.
>
> We appreciate that this might have been harder to understand based on our description and will try to rephrase the paragraph for clarity.
>
> ---------------------------
> Different terms used for ξ and typos
> "dynamical critical length ξ" in section 4.2. Later on the term "dynamical scale ξ" and "dynamical critical scale ξ" ….. "critical length χ"
>
> Thank you for spotting that we use the words “scale” and “length” interchangeably. We will adopt a single one to ensure clarity. You are right that in Figure 6 χ should have been ξ. This was a typo on our part and we will change it to  ξ.
>
> --------------------------
> “The equivalence between eq. 2 and the two parts of eq. 3 is not obvious. We'd appreciate if the authors would provide a proof of such. “
>
> The connection between Equation 2 and Equation 3 is very simple and we therefore believed it did not require a detailed proof, however, we’re happy to explain it in multiple steps. It is a Taylor expansion to the first order.
>
> The steps are as follows:
> The derivative of the loss L at image X1 with respect to the weight vector W is the gradient vector g1. If we look at the Taylor expansion of the change of loss due to a vector change of weights w, we obtain the dot product delta L = g1 dot w to the first order in the Taylor series. In particular, for a weight change induced by a small gradient step -epsilon*g1, we get delta L = - epsilon g1 dot g1. This is true as long as epsilon -> 0, which we take in the paper.
>
> --------------------------
> “The conclusions are self-evident.”
>
> While the self-evidence of our conclusions is a matter of subjective judgement, we do not believe that our results are in fact self-evident and our discussions with fellow researchers support this. It is hard to rebut this point, however, if you do not provide links to specific sources in literature.

---

> ### Author Response · Authors · 2019-11-15
> **Response to Review #4  2/2**
>
> ---------------------------
> “Hence, the reasoning is circular.”
>
> While we understand your point of view, i.e. that in general, when studying the stiffness between train and val examples and provided that both train and val losses go down with training time, you would expect stiffness to be high on average, this is true only *on average*. In our analyses, we go significantly beyond that along several dimensions and we will detail why we do not believe our reasoning is tautological:
>
> (a) While the on average the train-val stiffness can be expected to be high before overfitting, this does not explain the behavior of the val-val stiffness. The val-val stiffness is very different since not a single gradient step is taken with respect to a single validation image, yet the val-val stiffness mimics very closely the behavior of the train-val stiffness.
>
> (b) What you say is true *on average*, but does not address the stiffness behavior in detail, which is the bulk of our paper. We study the stiffness between examples based on their class membership and our results would not be predicted by simply stating that on average the stiffness must be high before overfitting. The same goes for the stiffness between different classes being negative but only slightly in magnitude.
>
> (c) The dependence of stiffness on the distance between images would not be predicted apriori, and neither would the decrease in the "dynamical critical scale ξ" with training time.
>
> ------------------------
> “Conclusion 7 is slightly different in that it observes that larger learning rates result in smaller ξ which, given the previous paragraph, we can rewrite into the statement "larger learning rates generalise better". This is a well known empirical observation and has been discussed thoroughly (e.g. on connection with flat and sharp minima or learning rate decay schedules). “
>
> We do not fully agree with your characterization of our results. It cannot simply be translated to "larger learning rates generalise better". In fact, the larger learning rates leading to functions that are influenced more locally by gradient updates (i.e. an input A will change the loss at input space positions within distance ξ, where ξ is smaller for large LRs) could be naively characterized as being exactly the opposite. A more local modification of the function could lead to *less* generalization. We therefore do not believe that your characterization is correct and while we do not have a good theoretical explanation yet, it certainly is not as simple a question as you made it sound.

---

> > ### Comment · AnonReviewer4 · 2019-11-15
> > **Response to the Rebuttal [1/2]**
> >
> > Thank you for your response and for addressing our comments. It is unfortunate that it was not given at a time that would still allow for a discussion.
> >
> > In general we found your responses to be compelling. It appears that we might be less knowledgeable about the area than previously thought. With that we would like to express less certainty in our rating.
> >
> > It appears to us that you agree that the phrasing is not ideal for a person that is not familiar with your work or deeply familiar with related work. As such we think that a revision of the text (in light of the reviewers comments) would be necessary. In the reviews you provided answers which in our opinion should be part of the text (with clear nomenclature).
> >
> > Another example we'd like to mention is how "dynamical critical length" is first defined in the Results section 4.2 whereas previous sections (like section 2.4) also mentions "dynamical critical length" without defining it. Ideally all terms can be fully understood before one reads the results section and the reader is guided towards a full understanding in a systematic way.
> >
> > In our previous response we have provided our own single sentence description of stiffness. Reading it again we noticed that we meant the difference in change but otherwise we assume you agree with our paraphrasing. We shall do that here for dynamical critical length as well for you to verify:
> >
> > Stiffness between two identical samples is 0.
> > Arguably, as the cosine distance of input samples grows as their absolute stiffness increases. (Though we don't recall that this is explicitly mentioned anywhere. Do you assume that stiffness is monotonic? We don't think that this is necessarily the case.)
> >
> > dynamical critical length (DCL): A learned linear estimate of the maximum distance between any two input points where where stiffness is (still) 0.
> >
> > It is not clear to us what aspect of this linear estimate is "dynamic" or "critical" but we understand that this might relate to established terminology that we are not aware of.
> >
> > We'll go through the comments now:
> >
> > 1.) Thank you for your clarification of section 4.4.
> >
> > So the question is wether a point x on a trajectory between two points in input space that goes from low stiffness to high stiffness correlates with a significant change in the prediction conditioned on that point x.
> >
> > We agree that this is unlikely to be the case and the value of this observation is not obvious to us.
> >
> > 2.) We welcome a more consistent nomenclature.
> >
> > 3.) Thanks for clarifying the connection between eq. 2 and eq.3. While simple, it is in our opinion not necessarily obvious at first glance and we recommend you add the necessary details for readers to follow your argument easily.
> >
> > 4.) Our "self-evident" statement is with regards to several sentences in the conclusion (1-6 in our initial review). We don't think reference to other work is necessary and instead picked four example sentences in the conclusion section that we consider obvious given the definition of stiffness:
> >
> > > Training-validation stiffness is directly related to the transfer of improvements on the training set to the validation set.
> > > We demonstrate the connection between stiffness and generalization and show that with the onset of overfitting to the training data, stiffness decreases and eventually reaches 0, where even gradient updates taken with respect images of a particular class stop benefiting other members of the same class.
> > > We find that in general gradient updates with respect to a member of a class help to improve loss on data points in the same class, i.e. that members of the same class have high stiffness with respect to each other.
> > > With the onset of overfitting, the stiffness between different classes regresses to 0, as does within-class stiffness.
> >
> >
> > 5.a) We agree with the authors that it is interesting that the val-val stiffness is mimicing the train-val stiffness that closely. That said, we don't find it necessarily surprising that the overfitting effect can be observed between different val-val samples. After the network overfitts to the samples in the train dataset there is no reason to believe that the network would implement a function that is class specific (since it overfits). As such it is not clear to us why one would expect that the stiffness (which depends on the implemented function) is going to be high for different validation samples, even if they are from the same class.
> >
> > 5.b/c) We tend to agree with the authors that there is more to it than our initial description might convey.
> >
> > 6.) We agree that our initial statement was overly simplistic. The paper states "The larger the learning rate, the smaller the stiff domains". Which you reiterate.

---

> > > ### Comment · AnonReviewer4 · 2019-11-15
> > > **Response to the Rebuttal [2/2]**
> > >
> > > We understand, please correct us if we are wrong, that with a larger learning rate the volume around an input vector (characterised by a small or zero change in loss), is smaller. We agree that the initial statement of ours is wrong. That said, the motivation and use of such an observation is not entirely clear to us. All that is mentioned in this regard is the the observation that larger learning rate leads to smaller stiffness. We don't have the expertise to judge ehy such an observation is valuable or interesting and abstain from judgment.
> > >
> > > Since you have not yet updated your submission we are unable to check several of our previous comments. We remain not fully convinced as we have hopefully made clear in this response to your rebuttal. Nevertheless, we consider our initial review unfit in light of the rebuttal and increase our score accordingly.

---

> > > > ### Author Response · Authors · 2019-11-15
> > > > **A response to the response to the Rebuttal [1/2]**
> > > >
> > > > Thank you for your quick and detailed reply. We appreciate that you are engaging in a discussion with us. While we primarily agree with your points on clarity of exposition to people unfamiliar with the subfield, we found several large points of misunderstanding in your response, primarily concerning the very definition of stiffness and the followup use of it in the definition of the critical dynamical scale. We will address those here.
> > > >
> > > > -----------------------------
> > > > Stiffness, as we define it in Eq. 4 and Eq. 5 (we look at two very related variants of it), is the *product between loss changes between two examples*. It is *not* the difference between them. If one example changes its loss by dL1, and the other one by dL2, stiffness ~ dL1 x dL2.
> > > >
> > > > We illustrate this in Figure 1. If both dL1 and dL2 have the same sign (e.g. if both L1 and L2 go down), the stiffness is high. If they do not care about each other, stiffness is 0. If they anti-correlate, i.e. the decrease of L1 increases L2, the stiffness is negative.
> > > >
> > > > Your point:
> > > > “Stiffness between two identical samples is 0. Arguably, as the cosine distance of input samples grows as their absolute stiffness increases.”
> > > >
> > > > This is certainly not true, from the definition. Between two identical examples, dL1 = dL2, and g1 = g2, therefore both definitions (Eq. 2 and 3) give you stiffness of 1, not 0. This is also clear from the illustrative Figure 1, where you can see that if X1 = X2, they have to change the loss the same, therefore they have stiffness of 1.
> > > >
> > > > We believe this might be a significant source of your confusion in the followup dynamical critical scale discussion.
> > > >
> > > > -----------------------------
> > > > Your point:
> > > > “Stiffness between two identical samples is 0. Arguably, as the cosine distance of input samples grows as their absolute stiffness increases. (Though we don't recall that this is explicitly mentioned anywhere. Do you assume that stiffness is monotonic? We don't think that this is necessarily the case.)
> > > >
> > > > dynamical critical length (DCL): A learned linear estimate of the maximum distance between any two input points where where stiffness is (still) 0.”
> > > >
> > > > We believe a part of this was cleared up by our previous point. Stiffness between two identical images X1 = X2 is 1. The farther X2 is from X1, the less stiff you would expect them to be. We measure this change of stiffness with distance between X1 and X2 and show our empirical results in Figures 5, 9 and 10. What you see there is the amount of stiffness between examples (y axis) going down from 1 as you increase the (cosine) distance between the two inputs X1 and X2 (x axis).
> > > >
> > > > We define the dynamical critical length to be the distance at which, for the first time, the stiffness between 2 examples reaches zero -- typically, the stiffness will be >0 for smaller distances (and be 1 as we approach the 0 distance).
> > > >
> > > > ---------------------------
> > > > Your point:
> > > > “ It is not clear to us what aspect of this linear estimate is "dynamic" or "critical" but we understand that this might relate to established terminology that we are not aware of. “
> > > >
> > > > The “critical” part is that this is the distance at which, for the first time (i.e. at no smaller distances typically) gradient updates will *not* induce correlated changes in loss between inputs X1 and X2. For smaller distances, the dL1 and dL2 and correlated on average. For larger distances, they are not strongly correlated. This dynamical critical length is “critical” in the sense that these two behaviours change there.
> > > >
> > > > The “dynamical” part is to distinguish it from the correlation length of the neural network outputs themselves. Let’s say the logits have a certain value at point X and for points at distance d from X there is no significant change in the logits. That would the normal (“static”) correlation length and it is dealt with in the spectral bias of NNs literature. In our case, we are interested in the correlations of *changes* of the loss (which is a proxy for the NN fn for us). In that sense, the length is “dynamical”. We use this term in order to distinguish it from the more usual correlation length, which characterizes the function outputs, and not their changes on gradient updates.

---

> > > > ### Author Response · Authors · 2019-11-15
> > > > **A response to the response to the Rebuttal [2/2]**
> > > >
> > > > ---------------------------
> > > > Thank you for your detail points. We believe many of them are related to the two misconceptions we tried to clarify above. We would like to respond to your last point:
> > > >
> > > > “….. with a larger learning rate the volume around an input vector (characterised by a small or zero change in loss), is smaller. We agree that the initial statement of ours is wrong. ….. motivation and use of such an observation is not entirely clear to us.”
> > > >
> > > > Thank you having a closer look and updating your review. We believe that there is a great value in understanding the specific details of functions we learn when we train deep neural networks on real data using gradient descent. Our work’s primary aim is to provide a new and interesting metric and characterize its behavior. We find stiffness and its input-space-distance behavior interesting, because it tells us about how local can the changes to the learned function be. If the dynamical critical length were really large, we could not learn any detailed class labeling on the input space. If it were too small, we would not be able to generalize. The fact that, according to this metric, the learning rate we choose leads to quite different functions learned, even though the more coarse-grained metrics such as loss do not seem to suggest so, is interesting to us. Our goal is to provide a piece of the puzzle in explaining and understanding how DNNs learn.

---

### Official Review · AnonReviewer5 · 2019-11-12
**Official Blind Review #5**

**Rating:** 3

**Review:**

This paper introduces “stiffness”, a new metric to characterize generalization in neural networks. Stiffness is a pretty simple concept and is relatively straightforward to compute. The authors evaluate this metric on standard datasets using two relatively small neural networks. On the whole, the paper is written clearly and explains its methodology in simple language.

I have a few observations:
1. The equivalence between equation 2 and equation 3 is mentioned in passing but no explanation is provided. Th equivalence is not clear so I would encourage the authors to provide a short proof.
2.  Since stiffness depends on the gradients obtained on points in the input space, which in turn depends on the loss, why would a practitioner training a neural network turn to stiffness to diagnose overfitting instead of just looking at the values of the training and validation losses? Indeed, the authors themselves say that a network has overfitted when training and validation losses diverge. The paper fails to motivate why stiffness is better than just looking at losses during training.
3. The authors mention “The train-val stiffness is directly related to generalization, as it corresponds to the amount of improvement on the training set transferring to the improvement of the validation set. ”. Typically, generalization is evaluated on a held out test set so I fail to understand what the authors mean by this statement. We would expect validation error to underestimate test error  so while they are related, train-val stiffness would not necessarily characterize generalization. It would be interesting to see a train-test stiffness graph to test the authors claim.
4. The paper fails to motivate the the utility of the concept of “Dynamical Critical distance”. Since the primary goal of  paper is to understand generalization, I would like the authors to clarify the motivation to study this quantity. What additional insight does this provide with respect to generalization?
5. The term “dynamical critical distance” is not used uniformly. For example, it is mentioned as “dynamical critical scale” in section 3.3 and “dynamical critical length” in section 4.2.
6. While the paper on the whole is written in a clear fashion, I found section 4.4 to be particularly confusing. The authors should consider rewriting that section to make it clearer.

In summary, the concept of stiffness seems to closely follow training and validation losses and any problem diagnosed using stiffness would therefore be also diagnosed via examining the loss values. This along with other concerns mentioned above mean that I cannot recommend this paper for publication.

**Experience Assessment:**

I have read many papers in this area.

**Review Assessment: Checking Correctness Of Derivations And Theory:**

I assessed the sensibility of the derivations and theory.

**Review Assessment: Checking Correctness Of Experiments:**

I assessed the sensibility of the experiments.

**Review Assessment: Thoroughness In Paper Reading:**

I read the paper at least twice and used my best judgement in assessing the paper.

---

> ### Author Response · Authors · 2019-11-15
> **Response to Review #5 1/2**
>
> Thank you for your review and comments. Your review was released to us only in the middle of the rebuttal period and we therefore didn't have the expected time to prepare a reaction. We would like to address several of the points your brought up.
>
> -------------------------
> “1. The equivalence between equation 2 and equation 3 is mentioned in passing but no explanation is provided.”
>
> The connection between Equation 2 and Equation 3 is very simple and we therefore believed it did not require a detailed proof, however, we’re happy to explain it in multiple steps. It is a Taylor expansion to the first order.
>
> The steps are as follows:
> The derivative of the loss L at image X1 with respect to the weight vector W is the gradient vector g1. If we look at the Taylor expansion of the change of loss due to a vector change of weights w, we obtain the dot product delta L = g1 dot w to the first order in the Taylor series. In particular, for a weight change induced by a small gradient step -epsilon*g1, we get delta L = - epsilon g1 dot g1. This is true as long as epsilon -> 0, which we take in the paper.
>
> -------------------------
> “2. Since stiffness depends on the gradients obtained on points in the input space, which in turn depends on the loss, why would a practitioner training a neural network turn to stiffness to diagnose overfitting instead of just looking at the values of the training and validation losses? “
>
> Stiffness as we defined it is related to the transfer of gains in performance from one input datapoint to another. It is defined by looking at the correlation between loss changes on different inputs, rather than the total loss at once. As such, it is a much finer metric than the total loss. However, you could look at loss changes on individual images and it would be equivalent -- that is exactly how we defined the concept of stiffness in the first place in Equations 1 - 5. Its connection to the gradient alignment is a mathematical consequence and it is a useful way to look at it, as it makes a direct connection to other works on gradient and Hessians in neural networks. However, you can definitely think about the correlation between changes in loss alone -- we call the particular product stiffness, since it geometrically relates to how easily “bendable” the learned NN function is.
>
> ----------------
> “3. The authors mention “The train-val stiffness is directly related to generalization, as it corresponds to the amount of improvement on the training set transferring to the improvement of the validation set. ”. Typically, generalization is evaluated on a held out test set so I fail to understand what the authors mean by this statement.”
>
> The misunderstanding here is between the validation set and the test set nomenclature. We used the term validation set to mean the held-out, never-trained-on dataset, and therefore used it to characterize generalization. You could easily call this the test set, since the important distinction wrt the train set is that not a single gradient step was ever taken wrt to a single image in it. We feel this point is only a matter of wording and we’ll try to make it clearer in the paper.
>
> ----------------
> “4. The paper fails to motivate the the utility of the concept of “Dynamical Critical distance”. Since the primary goal of paper is to understand generalization, I would like the authors to clarify the motivation to study this quantity. What additional insight does this provide with respect to generalization? “
>
> We firmly believe that our concept of dynamical critical length is of interest and well motivated, as it captures how localized changes to the learned NN function are when a gradient step with respect to a particular example is applied. It is very related to the rich literature on the spectral bias of neural networks. In our case, we study what could be seen as the dynamical equivalent of the spectral bias -- i.e. on which length scales in the input space do changes to the learned function stop affecting the rest of the function.
>
> The additional insight you are asking for is that this allows us to measure how localized changes to the learned function are. If you apply the gradient on input X1, the inputs X whose loss will change in a correlated manner are at a typical distance ξ or closer. We measure how this distance changes both with training time and the learning rate used. This gives us an additional diagnostic tool which is in turn directly useful in studying how independent the effects of gradient updates are between different images.
>
> ------------------------
> “5. The term “dynamical critical distance” is not used uniformly. …. The authors should consider rewriting that section to make it clearer. “
>
> Thank you for having a closer look. We will correct the typos and make sure to provide greater uniformity in the relevant subsection.

---

> > ### Comment · AnonReviewer5 · 2019-11-15
> > **Response to Rebuttal**
> >
> > Thank you for your response.
> >
> > 1. I still think this would be a valuable addition to the paper. A full proof or a proof sketch could be included in the appendix.
> > 2. I agree that stiffness is a finer metric than just looking at the losses. However, I do not understand the need for an entirely new metric to diagnose overfitting especially when it can be done by simple inspection of losses. Therefore, my question would be : what do we gain from this extra granularity for diagnosing overfitting? I understand that stiffness gives us (potentially) more information to work with, but how is that leveraged to improve our estimate of when overfitting occurs or understanding generalization in general.
> > 3. I was under the impression that network hyperparameters were chosen using the validation set. Therefore, I thought your statements regarding generalization may not hold. If this is not the case, I take back my objection.
> > 4.  Thank you for the clarification. On a closer reading, I now better understand the link you are trying to convey between the critical length and spectral bias towards learning low frequency functions early in training. This is a nice link to that topic.
> > 5. Thank you
> >
> > When I posted the first comment, I was under the impression that this paper is trying to introduce a new metric, hence, my questions on the marginal utility of a newer, finer metric when a simpler metric (looking at losses) does the same thing. Now, after looking at your comment and rereading the paper, we could interpret your work as trying to understand training dynamics and their connections to generalization by introducing a new quantity which serves as a proxy for aforementioned dynamics.
> >
> > Hence, the main concerns which remain for me are :
> > If this paper is about (1), we would expect a new metric to improve upon the shortcomings of an existing metric and, therefore, stiffness should be “better” (in some way) at diagnosing overfitting than plain inspection.
> > If the paper is about (2), apart from adding some empirical support to the idea of spectral bias, we do not seem to have an enhanced insight into the training dynamics. As an example, how does stiffness behave when the model *does not* generalize to the test set (or validation set according to the paper)?.
> >
> > Nevertheless, in light of your comment, I’ve updated my rating.

---

> ### Author Response · Authors · 2019-11-15
> **Response to Review #5 2/2**
>
> ------------------------
> “In summary, the concept of stiffness seems to closely follow training and validation losses and any problem diagnosed using stiffness would therefore be also diagnosed via examining the loss values. This along with other concerns mentioned above mean that I cannot recommend this paper for publication.”
>
> We do not agree with your characterization of our contribution. It is of course true that we could look at losses directly and it is indeed what we do -- we defined stiffness as the correlation between loss changes. That, however, is not the main contribution of our paper. We study how this concept depends on class membership, the stage of training, and the learning rate used for training. Those results could certainly not be inferred from the total loss and (according to us) deserve a closer look.

---

> > ### Comment · AnonReviewer5 · 2019-11-15
> > **Response to Rebuttal**
> >
> > I did not in any way intend to suggest that your contribution is limited only to that, but wanted to highlight	 the main thrust of my concerns and summarize them. Please see my comment below for an elaborate response.

---

### Decision · Program_Chairs · 2019-12-19

**Decision:**

Reject

**Comment:**

While there was some support for the ideas presented, the majority of reviewers felt that this submission is not ready for publication at ICLR in its present form.

Concerns raised include lack of sufficient motivation for the approach, and problems with clarity of the exposition.